# Successful Transvenous Extraction of Adult *Dirofilaria immitis* Parasites from a Naturally Infected Domestic Ferret (*Mustela putorius furo*)

**DOI:** 10.3390/ani14172474

**Published:** 2024-08-25

**Authors:** Eva Mohr-Peraza, Jorge Isidoro Matos, Sara Nieves García-Rodríguez, Alexis José Santana-González, Elena Carretón, José Alberto Montoya-Alonso

**Affiliations:** 1Internal Medicine, Faculty of Veterinary Medicine, Research Institute of Biomedical and Health Sciences (IUIBS), University of Las Palmas de Gran Canaria, 35016 Las Palmas de Gran Canaria, Spain; eva.mohr101@alu.ulpgc.es (E.M.-P.); saranieves.garcia@ulpgc.es (S.N.G.-R.); elena.carreton@ulpgc.es (E.C.); alberto.montoya@ulpgc.es (J.A.M.-A.); 2Anicura Veterinary Hospital, 35011 Las Palmas de Gran Canaria, Spain; alexis.santana@anicura.es

**Keywords:** parasitology, heartworm, veterinary, pathology, veterinary surgery, new companion animals, *Mustela putorius furo*

## Abstract

**Simple Summary:**

Heartworm disease is a severe and neglected cardiovascular condition in domestic ferrets. Isolated cases of natural infection have been previously reported. However, knowledge of the disease is still considered poor. An unsterilized 2-year-old domestic male ferret was diagnosed using a heartworm antigen test. The animal was asymptomatic and had a good physical appearance. A surgical transvenous extraction was performed, and two worms (one female and one male) were removed. The patient evolved favorably, and repeated diagnostic tests 35 days after surgery showed improvement in the parameters previously evaluated. This clinical case was the first in which it was possible to extract all the parasites from a ferret by endovascular therapy and improves the knowledge of the management of heartworm disease in ferrets.

**Abstract:**

Heartworm disease caused by *Dirofilaria immitis* is a serious and underdiagnosed cardiovascular condition in domestic ferrets. Hemodynamic changes caused by parasitization in ferrets cause a potentially fatal syndrome, but its clinical findings and treatment have not yet been standardized. The objective of this study was to describe the clinical case of a successful surgical extraction in a ferret infected by *D. immitis*. The patient was a 2-year-old, 1.5 kg asymptomatic male domestic ferret. The infection was diagnosed using a commercial test for the detection of *D. immitis* antigens. Subsequently, their clinical status was evaluated using serological and imaging diagnostic tests, and it was finally decided to perform surgical extraction of the adult worms. The ferret was anesthetized and placed in the left lateral decubitus position to perform a venotomy in the right jugular vein. Endoscopic extraction basket-shaped devices were used in the right atrial cavity under fluoroscopic guidance following the Seldinger endovascular surgery technique. With careful handling, two adult parasites were manually removed. A transthoracic echocardiogram performed after the procedure confirmed the absence of heartworms. The ferret recovered without complications and was discharged within 24 h. In the clinical review, 30 days after surgery, no notable alterations or symptoms were observed. This case report describes the first complete surgical removal of adult *D. immitis* parasites in a naturally infected ferret.

## 1. Introduction

Heartworm disease (*Dirofilaria immitis*) in ferrets is a poorly understood pathology, which puts this species at a disadvantage when it comes to the prevention, diagnosis, and treatment of this infection. Although it is well known that heartworm disease is a spreading disease, and awareness is high in companion animals such as dogs and cats, this is not the case for ferrets, and many epidemiological, pathophysiological, or clinical aspects are still barely studied.

The prevalence of heartworm disease in domestic ferrets has been scantily studied, but their susceptibility to *D. immitis* infection has been described to be similar to that of dogs [1], and natural infection appears to be related to the prevalence of dogs in endemic areas [1,2,3]. Ferrets experimentally inoculated with *D. immitis* larvae have been reported to have an overall infection rate approaching 100% [3].

Although parasite loads are relatively low, a few adult parasites can cause significant clinical disease and death [3]. Due to the large size of the parasites and the small diameter of the pulmonary arteries and right ventricles, most parasites lodge in the right ventricles and in the caudal and cranial veins, causing severe cardiovascular disease [4]. Although several case reports of heartworm disease in ferrets have been published, the factors that determine effective infection remain unknown [3,4,5,6,7,8,9].

Clinical signs commonly described in parasitized animals include cough, dyspnea, lethargy, anorexia, inability to exercise, cyanosis, pleural effusion, ascites, and hind limb paresis [3,5,6,9]. Clinical findings reported in previously published cases included severe cardiovascular disease characterized by pulmonary thromboembolism, severe pulmonary arteritis, eosinophilic or granulomatous pneumonia, and congestive heart failure [3,6,8]. Hematological abnormalities include monocytosis and anemia, and eosinophilia and neutrophilia have also been reported [5,9]. Hemoglobinuria has been reported on urinalysis [5], while one study described bilirubin in 83% of infected ferrets [4]. In addition, green-colored urine due to the presence of bilirubinuria has also been reported in a ferret with caval syndrome [3,6]. 

Diagnostic tests and procedures in ferrets are the same as those used in dogs and cats, although detection of microfilariae has been reported in only half of cases, usually due to the low number of live adults [1,3,5,6]. Antigen detection tests have been used to determine infection in ferrets, and ELISA-based antigen tests are considered effective for diagnosis in ferrets 5–6 months after infection [1], although sensitivity and specificity in ferrets are unknown. The radiological changes usually observed are cardiomegaly and pleural effusion, as well as an enlargement of the pulmonary arteries. Adult worms have been detected by echocardiography in experimentally infected animals from 5 months post infection [7], and several cases have also been described where hyperechogenic parallel lines were observed in the right ventricle, right atrium, and vena cava [3,6,8]. In addition, echocardiographic signs of pulmonary hypertension and right heart failure with severe atrial dilatation and retrograde venous congestion have been reported in a ferret [6].

Infection control is achieved by monthly administration of a routine heartworm preventative. This includes monthly administration of ivermectin (0.05 mg/kg, orally), milbemycin oxime (1.15–2.33 mg/kg, orally), selamectin (6–18 mg/kg, topically), or a combination of 10% imidacloprid and 1% moxidectin (0.4 mL/ferret, topically) [5,9,10].

There is currently no safe and effective adulticidal treatment for this species. Adulticidal treatment with melarsomine dihydrochloride has been used in ferrets with an unacceptable survival rate (<50%). Ferrets have a high risk of sudden death from thromboembolism due to parasite death, so its use is not recommended [9,11]. On the other hand, medical treatment with ivermectin and other macrocyclic lactones has also been found to be inappropriate in cases of parasitization [10]. Anecdotally, a study of four ferrets suggested that subcutaneous moxidectin (0.17 mg, every 6 months) may be an effective and safe adulticidal treatment [5]. In addition to pharmacological treatment, a single case of surgical removal has been reported, in which three adult parasites were removed from a ferret using a retrieval basket introduced through the right jugular vein, accessing the caudal vena cava and right atrium under fluoroscopic guidance. However, the authors were unable to remove all the parasites, leaving at least one adult worm in the right ventricle and the animal on chronic supportive care [6].

Therefore, given the limited experience with adulticidal therapies against *D. immitis* in ferrets, the aim of this study was to describe the first completed transvenous heartworm extraction in a naturally infected ferret and to discuss the clinical and diagnostic abnormalities associated with heartworm infection in the subject ferret.

## 2. Case Presentation

An unsterilized 2-year-old male domestic ferret was examined at the Veterinary Teaching Hospital of the University of Las Palmas de Gran Canaria (Spain). This ferret was participating in an awareness campaign on heartworm disease in this species, which consisted of the performance of a routine commercial immunochromatographic kit for the detection of *D. immitis* antigens (Urano test Dirofilaria^®^, Urano test Dirofilaria^®^, Urano Vet SL, Barcelona, Spain). The animal was asymptomatic, lived outdoors in a hyperendemic area for *D. immitis* [12], and was not receiving chemoprophylaxis against infection. The animal was positive for the antigen test. 

The animal did not exhibit aggressive behavior at any time and was not sedated or anesthetized for any test or clinical procedure. Physical examination revealed no signs of dehydration, and oral mucous membranes were slightly pale with a capillary refill time < 2 s. The body weight was 1.56 kg, and the body condition score was 3/5. A regular heart rhythm was observed with a frequency of 210 beats per minute and a respiratory rate of 45 breaths per minute, with no evidence of a pathological respiratory pattern. Thoracic auscultation revealed the absence of adventitious pulmonary sounds, but the presence of a moderate heart murmur (III/VI) was observed in the right caudal thoracic area, between the sixth and eighth intercostal spaces, compatible with the area of projection of the tricuspid valve. No changes were noted on abdominal palpation. A weak and regular femoral pulse was noted. There were also no changes in the extremities, with good proprioception reported [13]. 

The best-positioned, standard 2-view thoracic radiographs (dorsoventral and right lateral projections) were obtained (HFQ-600P, Bennett, New Columbia, PA, USA). In the dorsoventral projection, tortuosity and dilatation of the caudal pulmonary arteries were observed, with a ratio of 1.21 between the ninth rib and the left caudal pulmonary artery. In addition, a mildly inverted D-shaped cardiac silhouette was observed, compatible with right ventricular enlargement [14]. In addition, a globose heart with a Vertebral Heart Score (VHS) of 6.5 (reference range 5.23 to 5.47) was noted in the right lateral projection [15]. A mild diffuse and bilateral pulmonary broncho-interstitial pattern was also reported. An electrocardiographic study (GE MAC 800, General Electric, Boston, MA, USA) was performed with the animal in the dorsoventral position, showing a sinus rhythm of the order of 180 bpm with a cardiac axis of 73°. No changes in electrocardiographic parameters were reported during the study [13].

The ferret underwent a conventional transthoracic echocardiographic examination using a 12 MHz phased array ultrasound machine (Vivid IQ, General Electric, Boston, MA, USA). Echocardiographic findings included the presence of multiple parallel hyperechoic “equal signs” in the right atrium, indicating the presence of adult heartworms (Figure 1) [3,6]. Moderate dilatation of the pulmonary trunk, right atrium, and right ventricle was demonstrated with the presence of tricuspid regurgitation with a maximum velocity of 3.08 m/s (Table 1) [8].

Blood was collected from the jugular vein and a thick blood drop test was performed, which showed 13 microfilariae in 50 mcL. Laboratory biochemistry and complete blood count were performed, and no pathological changes were reported in any of the parameters evaluated (glucose, creatinine, urea, ratio between urea and creatinine, phosphorus, calcium, total proteins, albumin, globulins, ratio between albumin and globulins, aminotransferase, alkaline phosphatase, gamma-glutamyl transferase, total bilirubin, cholesterol, and D-dimer). An ultrasound-guided cystocentesis was performed, and the collected urine was slightly greenish in color. Urinalysis revealed a urine-specific gravity of 1.037 and a small amount of bilirubinuria (<0.3 mcg/mL) (IDEXX Catalyst TBIL Total Bilirubin, ME, USA) [13,16]. 

Due to the poor prognosis associated with standard medical treatment with melarsomine, in addition to the severe cardiovascular lesions associated with the infection, transvenous heartworm extraction was performed 24 h later at the Anicura Albea Veterinary Hospital (Las Palmas de Gran Canaria, Spain) with the owner’s consent. The procedure did not require the approval of an ethics committee as it was a nonexperimental clinical veterinary practice.

Prior to surgery, the ferret was hospitalized without medication, sedation, or fluid therapy. Anesthesia was induced and maintained with 2.5% isoflurane. The ferret was placed in a left lateral decubitus position and a conventional Seldinger technique was used through a venotomy in the right jugular vein [17,18]. A 4 French introducer was used and under fluoroscopic guidance (OEC One CFD, General Electric, Boston, MA, USA) a basket endoscopic retrieval device (Snare-System; 2.5–5 mm, Merit Medical, West Merit Parkway, South Jordan, UT, USA) was advanced into the right atrial cavity. Two heartworms (one male with conical and spirally coil, five pairs of preanal papillae, and six pairs of postanal papillae and one female with rounded caudal end and the vulvar opening located behind the junction of the esophagus and intestine) were removed manually with care (Figure 2). The jugular vein was then permanently ligated using vascular clips, and the incision was routinely sutured. A postoperative transthoracic echocardiogram documented the absence of heartworms. 

The main changes during the procedure were slight decreases in atrial pressures and multiple ventricular extrasystoles, mainly due to the extension of the entire extraction device into the right atrial cavity. The ferret recovered uneventfully and was discharged the following day and treated with meloxicam (0.1 mg/kg PO SID for 7 days), clopidogrel (3 mg/kg PO SID for 14 days), amoxicillin/clavulanic acid (20 mg/kg PO BID for 7 days), and selamectin (15 mg topically, monthly).

The patient was examined 35 days after the heartworm removal for further evaluation. The ferret was alert and showed no aggressive behavior. The animal showed good healing and resolution of the suture in the right cervical area. The owner reported no cardiorespiratory signs after discharge from the hospital. On physical examination, the animal manifested a weight gain of 1.63 kg (3.59 lb.) and remained with a body condition score of 3/5. A regular heart rate of 180 beats per minute and a regular respiratory rate of 39 breaths per minute were observed. There was no evidence of a pathological respiratory pattern or signs of dehydration. On thoracic auscultation, a slight murmur persisted in the projection area of the tricuspid valve (II/VI), while pulmonary auscultation was normal. The femoral pulse was strong, rhythmic, and symmetric [13].

A new radiological study was performed, and the dorsoventral projection showed a slight improvement in the tortuosity and dilatation of the caudal pulmonary arteries with a ratio of 1.08 between the ninth rib and the left caudal pulmonary artery. The inverted D-shaped cardiac morphology was also reported. The right lateral projection (Figure 3) confirmed the improvement of the pulmonary broncho-interstitial pattern and the reduction in the VHS to 5.5 [6,14,15]. An electrocardiographic study was repeated and showed a sinus rhythm of the order of 210 beats per minute with a cardiac axis of 74°, without pathological changes. The echocardiographic study performed during the evaluation of the animal showed the absence of parasites in the ventricles and vena cava, while a decrease in the velocity of tricuspid regurgitation (2.8 m/s) and a decrease in the diameter of the atrium, ventricle, and pulmonary trunk were demonstrated (Table 1) [8,13]. 

A blood sample was taken from the vena cava, and a new antigen detection test was performed with a negative result. A smear test was then performed, which showed no microfilariae. As before surgery, there were no changes in blood count and biochemical analysis. A new cystocentesis was performed that showed no bilirubin and a urine density of 1.034 [13,16].

Given the absence of symptoms and the improvement in clinical signs on imaging, no additional treatment was prescribed. Therefore, only monthly chemoprophylaxis (15 mg of topical selamectin) against heartworm disease was established.

## 3. Discussion

The diagnosis of heartworm disease in ferrets, as occurs in cats, presents serious difficulties, with the presence of symptoms, detection of antigens, and echocardiography being the main tools for its detection. Diagnosis is essential as the pathology has potentially fatal hemodynamic complications, and therapy is poorly developed in ferrets [1,3,5].

Similar to previous case reports, this ferret lived in a hyperendemic area, outdoors, and was not on prophylaxis against *D. immitis* infection, so the risk of disease was very high. Contrary to what has been described previously, the patient did not present with cardiorespiratory symptoms associated with infection, physiological parameters were also adequate, and the ferret had a good physical appearance. On examination, only the finding of a mild murmur in the area of the tricuspid valve projection was compatible with the presence of heartworm [6,8]. Hemodynamically, the patient’s functionality was adequate, contrasting with the cases described by Bradbury et al. (2010) [6] and Mihaylova (2018) [3]. In this instance, it is suspected that the larger size of the individual, along with the recent nature of the parasitism, prevented the adult heartworms from causing significant alterations at the time of diagnosis. This ferret had benefited from an awareness campaign about heartworm disease in this species, as the clinical signs are similar to those seen in dogs but often progress much more rapidly, making early diagnosis and treatment more urgent [7].

This ferret had two adult parasites, one male and one female, which facilitated diagnosis by microfilariae and antigen detection. Previous studies have shown that microfilariae detection tests are of little use in the diagnosis of ferrets [1]; however, the presence of microfilariae in this ferret demonstrated the possibility that infected ferrets may be reservoirs of the disease. Due to the rapid progression of the disease, heartworm must be diagnosed as early as possible and echocardiography has proven to be helpful in final diagnosis in previous cases [6,8,9]. In the current case, the presence of signs of right heart remodeling, in addition to findings compatible with adult parasites, was crucial for the identification of the infection.

Although the use of thoracic radiography reported the presence of signs of right cardiomegaly and dilation of the pulmonary arteries, it is true that they are not pathognomonic findings of heartworm disease [14]. However, these radiological findings should consider the presence of heartworm as a differential diagnosis, especially in hyperendemic areas, as previously described [4]. Likewise, the laboratory results of complete blood count and biochemistry did not experience changes before and after surgery and, unlike previous cases of heartworm in ferrets, anemia, leukocytosis, and eosinophilia were not observed [13,16]. However, the patient presented mild signs of bilirubinuria, as described in a similar case [6], which was also reversed after extraction surgery.

Transvenous heartworm extraction in ferrets had only been performed on one previous occasion; however, it was considered a viable option in the patient due to clinical stability. Unlike the present case, the previous surgical extraction was performed on a female ferret that presented severe cardiorespiratory symptoms, suffering from signs of right congestive heart failure, anemia, and leukocytosis [6]. Likewise, the same extraction technique was performed through a basket endoscopic retrieval device without serious adverse reactions or surgical complications observed. However, in the previous case, it was not possible to extract all the adult worms from the cardiac chambers; that female ferret experienced considerable symptomatic improvement after surgery, but cardiovascular damage persisted, and she was chronically medicated. The difference in the final result between these two procedures may be due to the different sizes of the individuals operated on, as well as the hemodynamic status of the patients.

## 4. Conclusions

This is the first reported case of successful complete extraction of all parasites from a ferret patient. After successful surgical removal, a marked improvement in the initially altered parameters was observed. Although the persistence of vascular damage typical of infection was evident 35 days later, the ferret had an excellent postsurgical quality of life, without the need for treatment. This highlights the need for awareness campaigns among ferret owners, which would allow early detection of this disease. Moreover, surgical removal by endovascular therapy may be a useful and effective tool in the treatment of ferrets with heartworm disease. There are no extensive studies, and extracting heartworms from such a small animal can be challenging, but it may currently be the best therapeutic option in this species, so it would be desirable to develop and standardize surgical techniques specifically for these animals, as has been carried out for dogs and is being performed on cats [17,18].

## Figures and Tables

**Figure 1 animals-14-02474-f001:**
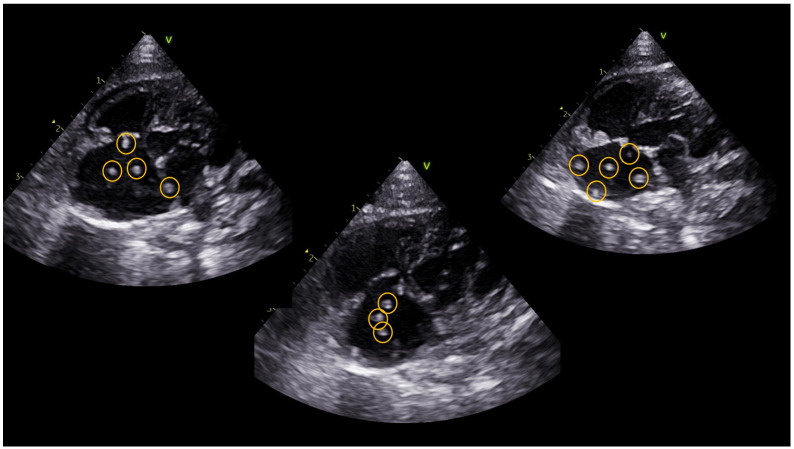
Visualization of multiple parallel hyperechoic “equal signs” in the right atrial cavity, consistent with adult *Dirofilaria immitis*, using a left parasternal 4-chamber apical echocardiographic view.

**Figure 2 animals-14-02474-f002:**
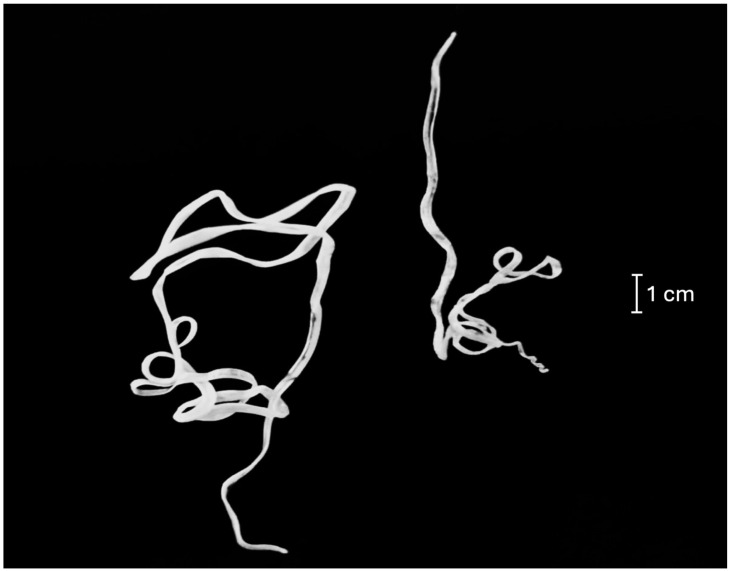
Adult *Dirofilaria immitis* parasites removed by endovascular therapy from a ferret undergoing surgery. Larger female worm (22 cm) and smaller male parasite (13 cm).

**Figure 3 animals-14-02474-f003:**
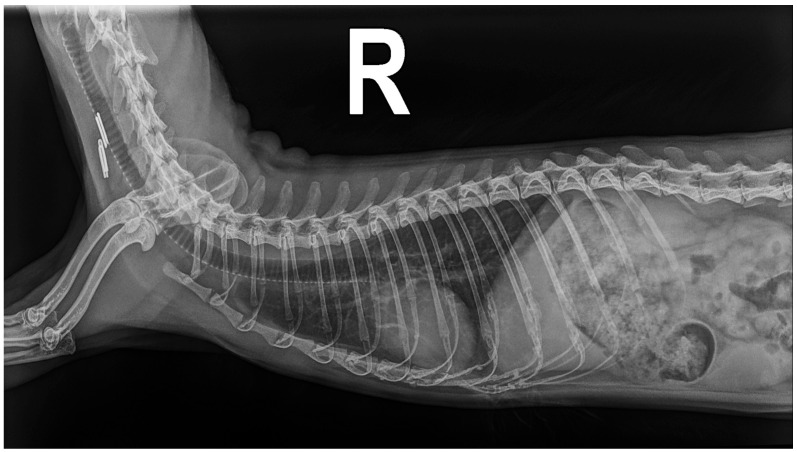
Radiological study in right lateral projection of the ferret 35 days after extraction of the adult parasites of *Dirofilaria immitis*.

**Table 1 animals-14-02474-t001:** Recorded echocardiographic measurements. RPADi: right pulmonary artery distensibility index; RVOT-FS: right ventricular outflow tract fractional shortening; RVEDAi: right ventricular end-diastolic area index; RAAi: right atrial area index; TRPG: tricuspid regurgitation pressure gradient; TAPSE: tricuspid annular plane systolic excursion. AT:ET: ratio of right ventricular acceleration time to ejection time; PT:Ao: pulmonary artery to aorta ratio; PV:PA: pulmonary vein to pulmonary artery ratio; E’: peak early diastolic endocardial velocity; A’: peak late diastolic endocardial velocity; S: peak systolic endocardial velocity; E:E’: ratio of early diastolic inflow to peak early diastolic endocardial velocity. The difference between each pair of observations before and after surgery is calculated as follows: Difference = {parameter value after surgery} − {parameter value before surgery}. Reference values for a healthy ferret performed by the same operator and with similar size, body weight, and sex as the patient undergoing surgery.

ECOCARDIOGRAPHIC MEASUREMENT	Before Surgery	After Surgery	Difference (after Value−before Value)	Healthy Ferret (Reference Values from Control Animal)
**RPADi**	28.6%	34.5%	5.9%	44.4%
**RVOT-FS**	33.3%	37.5%	4.2%	45.7%
**RVEDAi**	1.5 cm^2^	0.7 cm^2^	−0.8 cm^2^	0.54 cm^2^
**RAAi**	1.2 cm^2^	0.9 cm^2^	−0.3 cm^2^	0.36 cm^2^
**TRPG**	37.95 mmHg	31.36 mmHg	−6.59 mmHg	0
**TAPSE**	0.7	0.7	0	0.7
**AT:ET**	0.48	0.53	0.05	0.78
**PT:Ao**	1.4	1.2	−0.2	0.75
**PV:PA**	0.67	0.67	0	1.36
**E’**	0.08 m/s	0.09 m/s	−0.01 m/s	0.08 m/s
**A’**	0.11 m/s	0.04 m/s	−0.07 m/s	0.04 m/s
**S**	0.08 m/s	0.10 m/s	0.02 m/s	0.08 m/s
**E:E’**	6	4.5	−1.5	8

## Data Availability

The raw data supporting the conclusions of this article will be made available by the authors, without undue reservation.

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
