# Peer review of "Successful Transvenous Extraction of Adult Dirofilaria immitis Parasites from a Naturally Infected Domestic Ferret (Mustela putorius furo)"

_animals, 2024, doi:10.3390/ani14172474_

Round 1

Reviewer 1 Report

Comments and Suggestions for Authors

1)    Overall, the present case report is nothing special excluding the patient animal treat with the case.    However, this will be valued for so called exotic pet animal medical field and such veterinarians who are very busy.      

2)    Hence, the text is TOO long, they have to shorten that, especially both Introduction and result parts.       

3)    But, they extracted worms from the patient ferret, they DID NOT identify  the species.  Surely, that could be Dirofilaria, but the other genus, eg., Dipetalonema as well.    So, they have to add minimum information (description) about the nematodes extracted.

Reviewer 2 Report

Comments and Suggestions for Authors

• Line 20, replace “in” with “of”.

• In line 27, change 1.5-kg to 1.5 kg.

• Line 30 – it is suggested to replace the keywords “Dirofilaria immitis” and “Ferret” with others, as they already appear in the title of the work, increasing the indexing of the work.

• On line 84, between the words “ferret” and “topically”, insert a comma.

• On line 91, after the words “6 months)”, insert a comma.

• Lines 114 to 122 offer physiological parameters and semiological methodologies that must be referenced for the species. How do you know if your heart rate is normal or not, as well as your respiratory rate? Likewise, reference should be made to the classification of moderate heart murmur. Based on which study was this information concluded?

• In lines 123 and 124 it is reported that a radiographic examination was carried out. This technique should be referenced, allowing it to be replicated in similar situations. Is there any particularity in the execution of the technique in this species?

• Again, from lines 129 to 134 facts are reported that are not supported by other studies. How do you know if the electrocardiogram was normal? What literature was used to compare?

• In lines 145 and 146 it is reported that the ultrasound image is compatible with the presence of parasites. This finding must be supported by studies that report that, if this change is found, the presence of the parasite is confirmed.

• In table 1, the data can be compared statistically, whether they are different or not. Specific statistical tests for this can be applied. It is recommended that you consult a person specialized in statistics to provide better veracity to the findings. Additionally, a third column can be inserted with normal values ​​for a specimen, which can be obtained from other studies, or with a control animal being used. The proof of the need to carry out the test is in line 280, where it is reported that there was a great improvement in the parameters.

• In lines 164 and 165, acronyms are offered, without their meaning. Tests may have similar or different acronyms in different countries. As the magazine reaches very different populations, its meanings must be offered.

• In figure 2 there should be a size reference, such as a dash. Nowadays there are programs that allow you to do this, even if the photos are taken without an object that does this. If possible, it would greatly enrich the work. Furthermore, the image does not need to be so large, with the parasites small. The image could be improved.

• In lines 187 and 188, the authors report that a female and a male were extracted. It is requested that a reference be included that explains this (how to differentiate the genders in this species of parasite). This strengthens the work.

• In lines 207 to 211, facts of clinical examinations and their results are again reported, without theoretical basis, which are normal. They require referencing.

• In line 229 and 230 it is reported that a preventive treatment was instituted monthly, but it does not report which one or provide a reference that explains how to do it.

• The conclusion can be improved, with information offered in a more summarized and practical way, without repetitions.

Comments on the Quality of English Language

The authors did a good translation, using language that was easy and interesting to read and understand, but there were only gaps in understanding in the content of the text, which were mentioned in the comments to the authors.

Reviewer 3 Report

Comments and Suggestions for Authors

The MS “Successful transvenous extraction of adult Dirofilaria immitis parasites from a naturally infected domestic ferret (Mustela putorius furo)” provides a case report on ferrets, this data is scarce in scientific literature.

Some improvements are suggested to make clearer this MS:

INTRODUCTION

>Line 48-49: “The prevalence of heartworm disease in domestic ferrets is unknown, but their susceptibility to D. immitis infection has been described to be similar to that of dogs [1],..”

This is not a good reference as is from the year 1998, are there not any advances in prevalence of ferrets in the last 26 years? We still in the same situation?

>Line 56-57: “Although  several case reports of heartworm disease in ferrets have been published, the factors that  determine effective infection remain unknown.”

Need to include references.

>Line 59-68: Clinical signs commonly described in parasitised animals include cough, dyspnoea, lethargy, anorexia, inability to exercise, cyanosis, pleural effusion, ascites and hind limb  paresis [5, 4]. Clinical findings reported in previously published cases included severe 61 cardiovascular disease characterised by pulmonary thromboembolism, severe pulmonary  arteritis, eosinophilic or granulomatous pneumonia, and congestive heart failure [5]. Hae-63 matological abnormalities include monocytosis and anaemia, and eosinophilia and neutrophilia have also been reported [4]. Haemoglobinuria has been reported on urinalysis 65 [5], while one study described bilirubin in 83% of infected ferrets [4]. In addition, green  colour of the urine due to the presence of bilirubinuria has also been reported in a ferret  with caval syndrome [6].

Need to improve as only three papers are reported. Is no more studies being the 3 only published cases?, some of them are a review? Make it clear.

>Line 78-79 “In addition, signs of pulmonary hypertension and right heart failure with severe atrial dilatation and retrograde venous congestion have been reported in a ferret [8].”

Why is not this sentence together with clinical signs paragraph?

>Figure 1 is difficult to observe, have to be bigger.

>Figure 2 have to be bigger by removing the frame.

In the DISCUSSION several aspects need to be improved:

>Line 232: “The diagnosis of heartworm disease in ferrets presents serious difficulties”, why? Is not several available techniques?

>Line 240-241: On examination, only the finding of a mild murmur in the area of the tricuspid valve projection was compatible  with the presence of heartworm [6,8].

Some reason to explain not other clinical signs?

>Line 286-289: Extracting heartworms from such a small animal can be 286 challenging, but it may currently be the best therapeutic option in this species, so it would 287 be desirable to develop and standardise surgical techniques specifically for these animals, 288 as has been done in dogs and is being done in cats [15, 17].

Please provide mortality rates after extracting worms.

In CONCLUSIONS

Line 275-276: This clinical case highlights the risk of heartworm infection in ferrets living in hy-275 perendemic areas

Can you affirm this with a single ferret examined?

Other comments:

>the reference Mihaylova (2018) in Proceedings of 6th European Dirofilaria and Angiostrongylus Days should be included and discussed.

>The authors states that “This clinical case is the first where it is possible to extract all the parasites from a ferret by endovascular therapy”  but do not give the reasons, probably because was only two worms? In which part of heart was situated compared with Bradbury et al., 2010? As in my understanding is the same technique as previous published study of Bradbury et al., 2010.

Reference 3: Dirofilaria immitis have to be in italics

Reference 6 Heartworm Extraction Ferret With Caval Syndrome is not capital letters.

Reference 10: some words should not be capital letters

Round 2

Reviewer 2 Report

Comments and Suggestions for Authors

Congratulations on the corrections and the current work. The suggestions were accepted and greatly improved the article.